# Identification of B Cell Epitopes of Blo t 13 Allergen and Cross-Reactivity with Human Adipocytes and Heart Fatty Acid Binding Proteins

**DOI:** 10.3390/ijms20246107

**Published:** 2019-12-04

**Authors:** Marlon Múnera, Dalgys Martínez, Alexis Labrada, Luis Caraballo, Leonardo Puerta

**Affiliations:** 1Institute for Immunological Research, University of Cartagena, Cartagena 130000, Colombia; 2Centro Nacional de Biopreparados (BIOCEN), Bejucal 32600, Cuba

**Keywords:** FABP, IgE, IgG, house dust mites, cross-reactivity, *Blomia tropicalis*

## Abstract

Cross-reactivity between allergens and human proteins could have a clinical impact in allergic diseases. Blo t 13 is an allergen from the mite *Blomia tropicalis*, which belongs to the fatty acid binding protein (FABP) family and has structural homology with human FABPs. This work aimed to map B cell epitopes on Blo t 13 and to identify epitopes involved in cross-reactivity with human heart FABP (FABP3) and adipocyte FABP (FABP4). Sera from 25 patients with house dust mite (HDM) allergy that were sensitized to Blo t 13 were used for testing the reactivity of immunoglobulin E (IgE) and IgG to FABP. The epitope mapping of Blo t 13 was performed using overlapping peptides, and cross-reactivity between Blo t 13 and human FABP was analyzed using human sera and anti-Blo t 13 monoclonal antibodies. IgE antibodies to all FABPs were detected in 14/25 serum samples, and IgG was detected in 25/25 serum samples. The cross-reactivity of Blo t 13 was 42% with FABP3 and 48% with FABP4. Two IgE-binding regions were identified in Blo t 13; one between residues 54 and 72 (the main cross-reacting region) and another between residues 111 to 129. Our results suggest that exposure to the Blo t 13 allergen could induce an auto-reactive response to endogenous FABP in allergic patients sensitized to Blo t 13.

## 1. Introduction

An autoantigen is usually a protein that can be recognized by the immune system of patients suffering from a specific autoimmune response [1]. Several environmental allergens have the capacity to induce an autoimmune response meditated by immunoglobulin E (IgE) antibodies due to their homology with endogenous homologous proteins [2]. The clinical relevance of IgE responses to self-antigens remains largely unclear. However, their capacity to evoke allergic immediate-type reactions and to induce mediator release from basophils and mastocytes of sensitized individuals has been demonstrated. Based on these observations, it is reasonable to assume that IgE-mediated cross-linking of high-affinity IgE receptor (FcεRI) on effector cells induced by homologous autoantigens can elicit the same symptoms as those induced by environmental allergens, and this could explain exacerbations of chronic allergic diseases in the absence of external exposure [2]. Some autoantigens are homologous to environmental allergens, such as thioredoxins, manganese superoxide dismutase, profilin, and cyclophilin [2]. Immunoglobulin E-mediated autoimmunity has been studied in several diseases such as systemic lupus erythematosus, bullous pemphigoid, chronic urticaria and atopic eczema, which seem to be mediated, at least in part, by IgE autoantibodies [3]. Cross-reactivity between allergens and human antigens was explored by in vitro analysis combined with bioinformatic site-directed mutagenesis methods. For the cyclophilin family, Crameri et al. [4], using in silico analysis, predicted antigenic regions conserved among cyclophilins from *Malassezia sympodialis* (allergen: Mala s 6), *Aspergillus fumigatus* (allergen: Asp f 11), and from humans. These cyclophilins share 60% identity in their sequences. IgE antibodies from subjects sensitized to *M. sympodialis* and *A. fumigatus* reacted against cyclophilin from these fungi and against human cyclophilins. An inhibition assay confirmed the presence of cross-reactivity among environmental allergens and homologous endogenous antigens [4].

Similar approaches have been used to analyze cross-reactivity in other environmental allergens, such as thioredoxin and manganese superoxide dismutase (MnSOD) [5,6,7]. The *M. sympodialis* allergen Mala s 11 is homologous to MnSOD from *Homo sapiens*, with an identity grade of 50% in their amino acid sequences, as well as being homologous to *A. fumigatus* allergen (Asp f 6). Humoral and cell-mediated cross-reactivity between MnSOD from *H. sapiens* and *A. fumigatus* has been demonstrated. This may induce MnSOD to become an auto allergen in a subset of patients with atopic dermatitis and could evoke an autoimmune response, increasing clinical symptoms.

Thioredoxin autoreactive responses were studied using CD4+ T cell lines, and it was found that human and environmental thioredoxins can induce the proliferation of T cells committed to T-helper (Th)1, Th2, Th17, and Th22 subsets. Some cytokines, such as interferon (IFN)-γ, were detected and shown to have pathological roles, which might exacerbate atopic skin inflammation in sensitized patients through the activation of human thioredoxin specific T cells [8]. Several environmental allergens homologous to human proteins have roles in the development of autoallergy, such as profilins, bullous pemphigoid-180 (BP180), BP230, and serum albumins [9]. Other allergens with the capacity to induce an autoreactive response may exist. This could to help to explain the presence of several allergic symptoms in the absence of exposure to environmental allergens.

Here, we explored house dust mites (HDM) as a source of allergens with homology to human proteins and their implications for potential autoreactivity. HDM are important inducers of allergic responses [10]. Several allergens belonging to the FABP family have been identified in HDM. FABPs are intracellular proteins that play roles in the transportation and metabolism of long chain fatty acids [11]. The frequency of IgE reactivity in allergic patients has been reported to range from 13% to 23%; for example, Blo t 13 in *B. tropicalis*, Der f 13 in *Dermatophagoides farinae,* Der p 13 *in D. pteronyssinus,* Led d 13 in *Lepidoglyphus destructor* and Tyr p 13 in *Tyrophagus putrescentiae* [12,13].

The molecular modeling of Blo t 13 predicts an architecture consisting of 10 antiparallel β-strands forming two β-sheets surrounding an internal pocket or barrel structure and two short α-helices positioned at the end of the barrel [14]. In humans, the 13 members of the FABP family show predominant distributions in different tissues and organs. Some of them seem to be involved in the allergic inflammatory process in airways. The expression of FABP4 in airway epithelial cells is correlated with levels of Th2 cytokines (interleukin (IL)-4 and IL-5) and regulates the infiltration of eosinophils [15]. In endothelial cells, FABP4 is induced by vascular endothelial growth factor, a factor related to vascular remodeling in asthmatic airways [16]. FABP3 and FABP4 expression is largely restricted to macrophages and myeloid dendritic cells, cellular players in the asthmatic process. In macrophages, FABP4 regulates the activity of peroxisome proliferator-activated receptors (PPARγ) and nuclear factor kappa-light-chain-enhancer of activated B cells (NF-κB) signaling pathways, most likely by regulating the availability of key lipid signaling intermediaries [17].

In airway epithelial cell cultures, Der p 13 was shown to modulate the production of IL-8 and granulocyte macrophage colony-stimulating factor (GM-CSF) through toll-like receptor 2 (TLR2), myeloid differentiation primary response 88 (MyD88), NF-kB, and mitogen-activated protein kinases (MAPK)-dependent signaling pathways [13]. This suggests that FABP could contribute to the inflammatory process through innate immunity. Blo t 13 is homologous to human FABPs, having 46% amino acid identity and structural similarity with FABP3 and FABP4, which could support cross-reactivity. Molecular mimicry can induce an autoreactive response supported by cross-reactivity [3,18]. This turns the response into an allergic response, even in the absence of exposure to an environmental allergen [19]. In the present study, we cloned, produced, and analyzed the cross-reactivity among Blo t 13, FABP3, and FABP4. The epitope mapping assay identified two antigenic regions of Blo t 13, one of which was involved in IgE-mediated cross-reactivity between the allergen and human FABPs, which seems to explain the IgE-mediated autoreactivity found in sera from some HDM allergic patients.

## 2. Results

### 2.1. Blo t 13, FABP3, and FABP4 Share Two Conserved Regions

Multiple alignment revealed 46% identity in the amino acid sequences of Blo t 13, FABP4, and FABP3 (Figure 1) with two highly conserved regions. One region spans residues 10 to 28 located in α-helix I, and the other region spans residues 57 to 74 located in β-strands βC cand βD. Twenty-nine out of the 40 conserved residues are surface exposed, and most of them are charged. When Blo t 13 is superimposed on the FABP3 and FABP4 structures (Figure 2), the positions of 130 equivalent α carbons in the three molecules can be aligned with a Root Median Square Deviation (RMSD) of 0.8 Å.

### 2.2. IgE and IgG Reactivity to FABP3 and FABP4 in Allergic Patients Sensitized to Blo t 13

The frequency of IgE reactivity to FABP3 was 56% and 55% to FABP4. Fourteen out of the 25 serum samples (56%) reacted with all FABPs (Figure 3, Table 1). Some sera reacted to one FABP only, e.g., 10, 18, and 21. All serum samples from allergic and control subjects had positive IgG and IgG4 levels to at least one FABP. The IgG levels to Blo t 13 were significantly higher than those to FABP3 and FABP4 (*p* < 0.05) (Figure 4).

### 2.3. Blo t 13 Cross-reacts with Human FABPs

The inhibition of IgE and IgG reactivity to Blo t 13 by FABP3 and FABP4 showed a dose response curve with the highest inhibition at a concentration of 100 µg/mL, indicating a moderate to high cross-reactivity (Figure 5A–D). No inhibition was observed with bovine serum albumin (BSA). Among the monoclonal antibodies (mAbs), the 5G3 mAb had higher reactivity to FABPs (Figure 6A). The reactivity of 5G3 was focused on peptides 4, 5, and 6 (Figure 6B).

### 2.4. Epitope Mapping Reveals Two IgE Binding Regions in Blo t 13

Epitope mapping identified two IgE binding regions in Blo t 13 (Figure 7A,B). The first region spans residues 54 to 72, and the second spans residues 111 to 129. Peptide 5 overlaps with the first identified IgE binding region, which is the most conservative region among the FABPs (Figure 1A). IgE binding regions are located on the β strands βD, βE, βJ and βI (Figure 1B,D). According to structural comparisons, they are in areas with high homology between Blo t 13 and human FABPs. The inhibition assay with different concentrations of peptides 4, 5, and 9 revealed a dose–response relationship, supporting the presence of IgE epitopes in these regions (Figure 7C). In order to confirm whether residues covered by peptide 5 are involved in the cross-reactivity, we performed an ELISA inhibition assay. Peptide 5 inhibited IgE reactivity to FABP3 and FABP4 from the serum pool at 60% (Figure 7D).

## 3. Discussion

Cross-reactivity among environmental allergens and human antigens has been suggested as a mechanism to trigger IgE mediated autoreactivity and to promote exacerbations of symptoms in allergic diseases [3,20]. Some allergic subjects exhibited IgE reactivity to Blo t 13, FABP3 and FABP4, and the inhibition assay demonstrated cross-reactivity among them. This was supported by using an anti-Blo t 13 monoclonal antibody that reacted against FABP. Epitope mapping identified two B cell epitopes on Blo t 13, located in residue positions 54 to 72 and 111 to 129. The first one was involved in cross-reactivity between Blo t 13 and human FABPs. This is the first experimental report of the identification of a cross-reactive antigenic region of Blo t 13, which could help to identify the regions involved in the autoreactive response due to molecular mimicry.

Allergens of group 13 of HDM could represent a group of allergens that could induce sensitization to homologous endogenous FABPs. Other allergens from various sources and with different biological functions display autosensitization capacity [3,5]. Molecular mimicry describes the occurrence and pathogenic consequences of common B or T cell reactive epitopes between microorganisms or environmental agents and the host. Molecular mimicry occurs at different levels, including complete identity or homology at the protein level, similarity at the amino acid sequences, and structural similarity [21]. Also, molecular mimicry between B cell epitopes is crucial for the cross-linking of FcεRI receptors by IgE antibodies on effector cells that could elicit the same symptoms as those induced by environmental allergens, and this could explain exacerbations of chronic allergic diseases in the absence of external exposure [2]. Comparison of the Blo t 13, FABP3, and FABP4 structures revealed that, despite a moderate sequence identity (46%), the overall fold of the core region is quite similar with an RMSD score of 0.8, suggesting potential cross-reactivity between them.

In previous work, cross-reactivity between an allergen from *D. farinae* (Der f 13) and eight human FABPs were explored by Chew et al. [22]. However, no cross-reactivity was detected, although Der f 13 and human FABPs have structural homology and share 58% identity in the amino acid sequences [22]. In contrast, for human and environmental profilins with only 34% sequence identity, both cross-reactivity and the capacity to induce auto-reactive skin responses has been reported [23,24]. There are two possible reasons explaining why Chew et al. failed to find cross-reactivity among Der f 13 and human FABPs. First, they used a limited amount of serum samples, which is an important factor to consider in presence of an event that occurs at very low frequency. Second, the genetic backgrounds of Asian populations are quite different to our population.

Although FABPs have been reported in several organisms and allergenic sources [25], their characterization is limited. Here, we provide the first evidence of cross-reactivity between FABP from HDM and humans supported by the identification of a common B cell epitope shared among Blo t 13, FABP3 and FABP4. Other studies have identified epitopes in FABP from *Schistosoma japonicum* and *Echinococcus granulosus* [26,27] by using methodologies based on synthetic peptides, however, none of them were reported as allergens. For epitope mapping of Blo t 13, an inhibition assay was performed with synthetic peptides covering the full amino acid sequence of this allergen. This method was reported as useful for mapping immunoglobulin IgE epitopes in allergens from HDM [28]. Importantly, the presence of the cross-reactive antigenic region was confirmed in a dose–response assay using the peptides with high levels of inhibition in the initial assay. In this manner, we discard the induction of false positives by nonspecific reactions to synthetic peptides used on the solid phase.

Autoreactive responses have been suggested to play a pivotal role in the development of allergy and asthma [29,30]. Although FABPs are intracellular, we hypothesize that inflammation in allergy responses could contribute to their release and accessibility to be recognized by antibodies raised against environmental FABP, contributing to the perpetuation of inflammatory and autoreactive responses. To explore this possibility, it is mandatory to perform other studies, which include T cell epitope analyses, and use of a murine model to test the capacity of Blo t 13 to induce an autoreactive response to endogenous FABP.

Blo t 13 is the first HDM allergen to which a cross-reactivity with human proteins has been demonstrated; therefore, the probability that an autoreactive response to homologous proteins such as FABP3 and FABP4 could be induced during the course of an allergen immunotherapy (AIT) with HDM extract deserves evaluation. In some studies, immunologic diseases such as autoimmune diseases are considered a relative contraindication for AIT [31]. However, in a large nationwide study analyzing data over a decade, patients treated with AIT had lower incidence of autoimmunity compared to those on conventional treatment (nasal steroids and oral antihistamines) [32]. In addition, Bozek et al. [33] found in a long-term observational study that specific immunotherapy with HDM extract is indeed safe and did not find evidence to support that this immunotherapy is associated with increased risk of the development of autoimmune disease.

In conclusion, our study provides evidence of cross-reactivity between group 13 HDM allergens and human FABP3 and FABP4. The identification of a common epitope in these proteins suggest that they might participate in a potential autoreactive immune response, which could have clinical implications in allergic individuals.

## 4. Materials and Methods

### 4.1. Recombinant Proteins

The nucleotide sequences of Blo t 13.0101 (GenBank: AAC80579.1), FABP3 (CAA39889.1) and FABP4 (EAW87092.1) were inserted into the expression vector pET45b+ (Genscript. Nanjing, China) to transform the *Escherichia coli* strain BL21 (DE3). The strain was grown in Luria–Bertani (LB) medium containing 100 mg/L ampicillin at 37 °C to an optical density (O.D.) of 0.4 at 600 nm. Protein expression was induced by the addition of 1 mM isopropyl-β-D–thio-galacto-pyranoside, and the culture was incubated for 4 h at 37 °C. Cells were then harvested by centrifugation at 6500 rpm at 4 °C for 15 min. The cell pellets were solubilized in 8 mol/L urea, 0.1 mol/L NaH_2_PO_4_, and 0.01 mol/L Tris-HCl at pH 8.0 and pulsed by ultrasound four times at 100 MHz on ice and then incubated by continuous rotation for 3 h. Insoluble material was removed by centrifugation. The supernatant containing the recombinant protein was applied to nickel–nitrilotriacetic acid–agarose (Qiagen. Hilden, Germany) and purified in hybrid conditions, as indicated by the supplier. The proteins were eluted with native elution buffer (50 mM NaH_2_PO_4_, 0.5 M NaCl, and 250 mM imidazole at pH 8.0). Fractions containing eluted proteins were pooled and dialyzed against 50 mM NaH_2_PO_4_ and 0.5 M NaCl (pH 8.0). The protein concentration was determined by Bradford assay (Bio-Rad Laboratories, San Francisco, CA, US). The purity and integrity of all proteins were assessed by SDS-PAGE and subsequent staining with Coomassie Brilliant Blue R-250 (Bio-Rad Laboratories).

### 4.2. Serum Samples

Sera were obtained from the Institute for Immunological Research (University of Cartagena, Cartagena, Colombia). All patients were skin prick tested with a battery of common standardized inhalant allergens. Individuals who developed a wheal larger than 3 mm in diameter to extracts of *B. tropicalis* and *D. pteronyssinus* were considered as having a skin prick test (SPT) result. To evaluate the IgE and IgG binding capacity of purified Blo t 13, FABP3, and FABP4, twenty-five patients with allergy to HDM who were sensitized to Blo t 13 were included (Table 1). In addition, a serum pool was prepared by mixing equal volumes of four positive selected serum samples from these patients. Written consent was obtained from the participants in the study. The Bioethics Committee of University of Cartagena, Colombia, approved this study (date of approval: 10 July 2010; project code: 110752128386).

### 4.3. Monoclonal Antibody Reactivity to Blo t 13 and Human FABPs

The reactivity of anti-Blo t 13 mAbs, named 5G3, 5H11 and 6D6 [34], was assessed against FABP3 and FABP4 by ELISA. Microtiter wells (Immulon-4, Dynatech, Chantilly, VA, USA) were incubated with 0.5 μg/100 μL of recombinant FABP overnight at 4 °C. After several washes, wells were blocked with 100 μL of 1% BSA and 0.02% sodium azide in phosphate buffered saline with Tween-20 (PBS-T) for 3 h at room temperature in a wet chamber and incubated with 0.5 µg/100 μL of mAb for 2 h at room temperature. After several washes, 100 μL of alkaline phosphatase-conjugated anti mouse-IgG (Sigma, ref. A3562, San Luis, MO, USA), diluted 1:10,000 in buffer containing 0.05 M of Tris (pH 8.0), 1% BSA, 1 mM MgCl_2_, and 0.02% sodium azide was added to the wells with incubation for 2 h at room temperature. After the final wash, a colorimetric reaction was developed by adding paranitrophenyl phosphate and incubating for 30 min at room temperature. Then, the absorbance at 405 nm was determined using a spectrophotometer (Spectra MAX 250, Molecular Device, Sunnyvale, CA, USA). Results are expressed as the O.D.

### 4.4. Escherichia coli Lysate Preparation for Serum Absorption

One colony of *E. coli* (BL21) DE3 Start cells from LB culture on petri dishes was inoculated in 1 L of LB medium. After 24 h of incubation at 37 °C and 250 rpm, the culture was centrifuged at 6500 rpm at 4 °C, and the cell bottom was suspended in lysis buffer (2 M Tris HCl (pH 8.0), 0.2% sodium azide) and sonicated. The lysate was centrifuged at 6500 rpm, and the supernatant was recovered. For specific detection of IgG and IgG4, serum samples were pre-adsorbed with *E. coli* lysate by mixing 500 μL of lysate with 1000 μL of serum and incubated at 4 °C with rotation overnight.

### 4.5. Determination of Serum Antibodies Against FABP and ELISA Inhibition

IgE reactivity was determined by ELISA; microtiter wells (Immulon-4, Dynatech, Chantilly, VA, USA) were incubated with 0.5 μg/100 μL of Blo t 13, FABP3, or FABP4 overnight at room temperature. After three washes with PBS-T, the wells were incubated with 100 μL of blocking buffer (1% BSA and 0.02% sodium azide in PBS-T) for 3 h at room temperature in a wet chamber. Then, they were incubated with 100 µL of serum samples diluted 1:5 overnight at room temperature, and after washing, they were incubated with 100 µL of alkaline phosphatase conjugated anti-IgE (Sigma A3525) diluted 1: 500 in 0.05 M Tris and 1 mM MgCl_2_ for 2 h at room temperature. After the final wash, 100 μL of paranitrophenyl phosphate was added, and the wells were incubated at room temperature for 30 min. Absorbance at 405 nm was determined using a spectrophotometer (Spectra MAX 250, Molecular Device, Sunnyvale, CA, USA). Results are expressed as the O.D. and all experiments were performed in duplicate. Sera from three non-allergic individuals were used as negative controls. The mean values from these samples plus two standard deviations were used as the cut off values for positive IgE levels.

For IgG and IgG4 determination, serum samples were pre-absorbed with *E. coli* lysate and diluted 1:100 and 1:50, respectively. The secondary antibody (conjugated) anti-IgG (Sigma A1543) was diluted in 1:10,000, and phosphatase conjugated anti-IgG4 alkaline (Pharmigen, San Jose, CA, USA), Cat N° 555880) diluted 1:500 was added, and an ELISA was performed as described above.

Cross-reactivity between Blo t 13 and FABP3 and FABP4 was analyzed by ELISA inhibition; one volume of the serum pool was adsorbed with one volume of increasing concentrations of an inhibitor (Blo t 13, FABP3, or FABP4) for 10 h at 4 °C. Then, 100 µL was loaded into wells coated with the relevant antigens and incubated overnight at room temperature in a wet chamber. After three washes with PBS-T, bound antibodies were measured by ELISA as described above. The percentage of inhibition was calculated as follows: (O.D. sample without inhibitor – O.D. sample with inhibitor / O.D. sample without inhibitor) × 100.

### 4.6. B Cell Epitope Mapping of Blo t 13

For epitope mapping, ten overlapping peptides representing amino acids 1–20, 14–33, 27–46, 40–59, 53–72, 66–85, 79–97, 92–101, 105–124 and 116–130 of the Blo t 13 sequence (Uniprot: Q17284) were prepared by solid phase synthesis with an automated peptide synthesizer, purified by high-performance liquid chromatography, and analyzed by mass spectrometry (Genscript, Nanjing, China). Lyophilized peptides were reconstituted to 10 mg/mL in 2.5% dimethyl sulphoxide and diluted in PBS.

Epitope mapping was performed as indicated elsewhere [28] with modifications. One hundred microliters of seven individual sera samples from allergic patients sensitized to Blo t 13 diluted 1:2 was absorbed with 150 µg/100 µL of each peptide for 5 h at room temperature. BSA (150 µg/100 µL) and Blo t 13 (10 µg/100 µL) were used as negative and positive controls, respectively. Peptides showing a significant level of inhibition were then tested at different concentrations to observe the dose–response relationship. The degree of inhibition was expressed as the CRI using the following formula: (peptide inhibition % / homologous inhibition %) × 100. CRI values below 25% were considered as representing no inhibition and those between 25% and 50% as having a low degree of inhibition. Two serum samples from non-allergic subjects with negative IgE to Blo t 13 were used as controls. All experiments were performed in duplicate.

To map the epitopes recognized by 5G3 mAb, a modified protocol was used: 0.1 µg/100 µL of 5G3 mAb was pre-adsorbed with 150 µg/100 µL of each peptide and incubated with Blo t 13 on the solid phase.

### 4.7. Cross-inhibition Assays by ELISA—Cross-Reactive Epitopes

In order to identify the cross-reactive epitope recognized by 5G3 mAb, an ELISA inhibition reaction was performed as follows: the mAb (0.5 µg/100 µL) was incubated with each peptide (150 µg/100 µL) or Blo t 13 (10 µg/100 µL) in a volume ratio 1:1 for 5 h at 4 °C, and then 100 μL of this mix was loaded into wells coated with Blo t 13 and incubated for 2 h. After several washes, wells were incubated with 100 µL of anti-mouse-IgG (conjugated) (Sigma A3562) diluted 1:10,000 for 3 h. After the final wash, wells were incubated with 100 µL of paranitrophenyl phosphate (Sigma N7653) at room temperature for 30 min. The reaction was stopped with 100 µL of NaOH (3 N) and the absorbance was determined at 405 nm by using a spectrophotometer (Spectra MAX 250, Molecular Device, Sunnyvale, CA, USA). The degree of inhibition was expressed as the CRI. In addition, to test whether peptide 5 recognized by 5G3 did contain an IgE cross-reactive epitope, 100 µL of the serum pool diluted 1:4 was incubated with each peptide (150 µg/100 µL) or Blo t 13 (10 µg/100 µL) for 5 h at room temperature. Wells coated with Blo t 13, FABP3, or FABP4 were incubated with this mix overnight at room temperature. An ELISA was performed as indicated above. The result was expressed as the CRI. The allergen Lit v 1 from *Litopenaus vannamei* was used as the negative control. All experiments were performed in duplicate.

### 4.8. Sequence Alignment and Structural Modeling

The amino acid sequences of Blo t 13 (accession number Q17284), Der f 13 (Q1M2P5), FABP3 (P05413) and FABP4 (P15090) were retrieved from the UNIPROT database [35]. The amino acid sequences were aligned using the bioinformatic tool “PRALINE” from the web server IBIVU [36]. Homology-based modeling of the Blo t 13 structure was done in the SWISS-MODEL server using 3D templates of Der f 13 (Protein Data Bank ID: 2A0A) [37]. Models were refined in Deep-View (energy minimization and rotamer replacements). Their quality was evaluated by several tools, including Ramachandran plots, the Qualitative Model Energy Analysis (QMEAN4) index, and energy values (GROMOS96 force field).

For structural comparison, the 3D structures of FABP3 (PDB ID: 3RSW) and FABP4 (PDB ID: 3RZY) were retrieved from the Protein Data Bank and submitted to Chimera UCSF software using the 3D model of Blo t 13 as a query for the RMSD index calculation [38,39]. Solvent-accessible surface areas of the Blo t 13 structure for residues predicted to be involved in antigenic regions were calculated with the web tool Accessible Surface Area (ASA) view; a value >0.25 was considered to represent exposure [40].

### 4.9. Statistical Analysis

The comparison of antibody levels was conducted using the non-parametric Mann–Whitney U test. All statistical analyses were two-tailed, and the significance was set at *p* < 0.05. Analyses were performed using International Business Machines (IBM) Statistics Statically Product and Service Solutions (SPSS v20, IBM Corp., Armonk, NY, USA) and GraphPad Prism (GraphPad Software, San Diego, CA, USA).

## Figures and Tables

**Figure 1 ijms-20-06107-f001:**
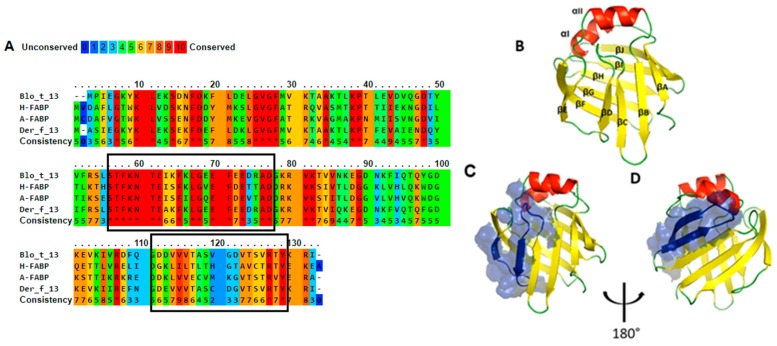
Homology analysis. (**A**). Multiple alignment, fatty acid binding proteins (FABPs) share 46% identity among amino acid sequences, the boxes indicate antigenic regions mapped on Blo t 13 (**B**). Three-dimensional (3D) model of Blo t 13. (**C**) and (**D**). Representation of first and second epitope on the 3D model, respectively.

**Figure 2 ijms-20-06107-f002:**
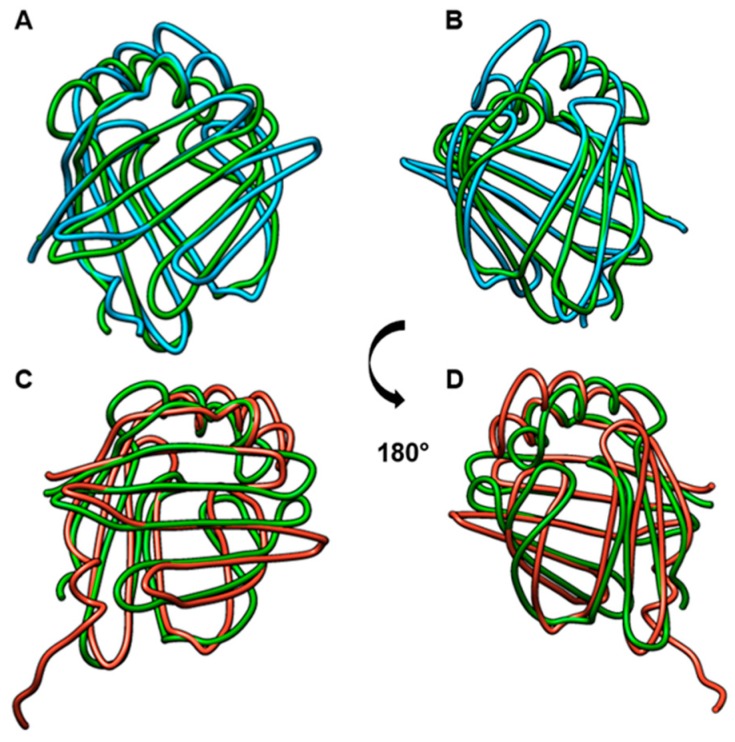
Comparison of FABP structures. (**A**) and (**B**): Comparison of Blo t 13 (green) with FABP3 (blue). (**C**) and (**D**). Comparison of Blo t 13 (green) with FABP4 (red). RMSD = 0.8.

**Figure 3 ijms-20-06107-f003:**
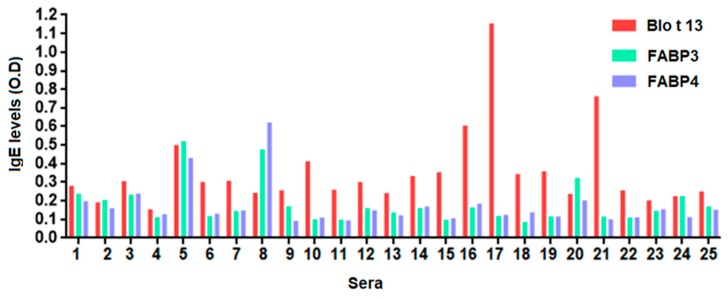
Specific immunoglobulin E (IgE) levels to FABPs using serum from allergic subjects sensitized to Blo t 13. Fourteen serum samples (56%) exhibited IgE reactivity to Blo t 13, FABP3 and FABP4. An optical density (O.D.) value equal to or greater than 0.12 was considered positive.

**Figure 4 ijms-20-06107-f004:**
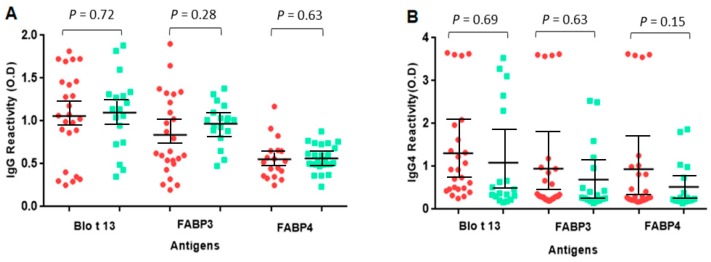
Specific IgG and IgG4 levels to FABPs. The serum levels of IgG (**A**) and IgG4 (**B**) between allergic and control groups did not show significant differences (*p* > 0.05). Red circles represent allergic patients and green squares represent controls. Non-parametric Mann–Whitney U test was performed. Vertical bars represent standard deviation.

**Figure 5 ijms-20-06107-f005:**
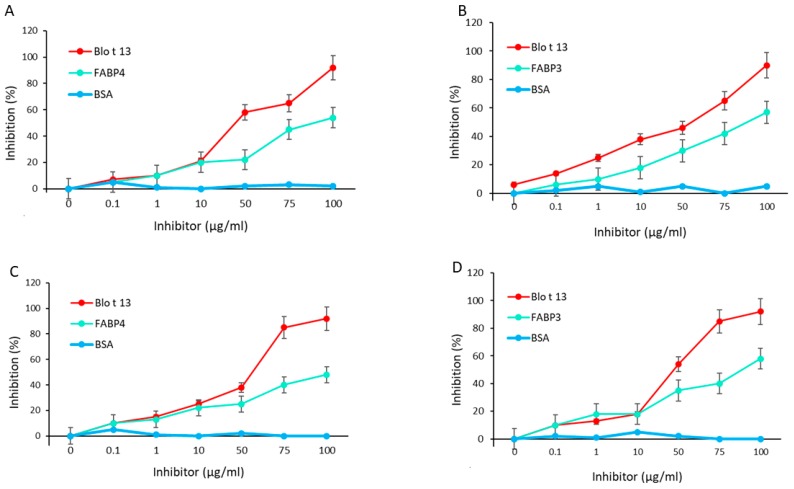
IgE and IgG cross-reactivity assays using Blo t 13 on solid phase. (**A**) FABP4 inhibited IgE reactivity to Blo t 13 at 55%. (**B**) FABP3 inhibited IgE reactivity to Blo t 13 at 62%. (**C**) FABP4 inhibited IgG reactivity to Blo t 13 at 49.2%. (**D**) FABP3 inhibited IgG reactivity to Blo t 13 at 54.7%. BSA was used as a negative control.

**Figure 6 ijms-20-06107-f006:**
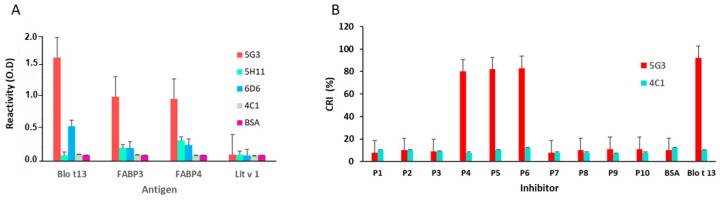
ELISA results using monoclonal antibodies. (**A**) Reactivity of anti Blo t 13 monoclonal antibodies (5G3, 5H11 and 6D6) and anti Der p 1 (4C1) to FABPs and Lit v 1 allergen. (**B**) Reactivity of 5G3 and 4C1 to each peptide from Blo t 13 indicated by CRI. Peptides 4, 5 and 6 showed the higher reactivity with the 5G3. The 4C1 mAb and Lit v 1 allergen were used as controls.

**Figure 7 ijms-20-06107-f007:**
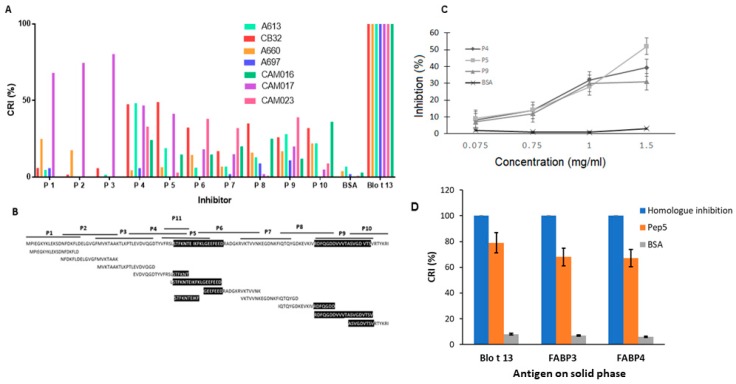
Epitope mapping results. (**A**) Serum samples from seven allergic patients were preabsorbed with each of the 11 overlapping peptides, Blo t 13 or BSA and incubated in wells coated with Blo t 13. (**B**) Amino acid sequence of overlapping peptides of Blo t 13. The two antigenic regions identified in epitope mapping experiments are shaded. (**C**) Dose–response curves for inhibition of IgE binding to Blo t 13. Peptides 4, 5, and 9 inhibited reactivity by 39%, 52% and 34%, respectively. BSA was used as a negative control. (**D**) IgE binding inhibition assays. Peptide 5 inhibited IgE reactivity to Blo t 13, FABP3 and FABP4. Cross Reactivity Index (CRI).

**Table 1 ijms-20-06107-t001:** Patients selected for the study.

Patient	Sex	Prick Test > 3 mm	IgE Levels (O.D.) Median		
			**Blo t 13**	**FABP3**	**FABP4**
1	M	Bt, Dp	0.28	0.20	0.24
2	M	Bt, Dp	0.19	0.16	0.20
3	F	Bt, Dp	0.30	0.24	0.23
4	F	Bt, Dp	0.16	0.13	0.11
5	F	Bt, Dp	0.30	0.13	0.12
6	F	Bt, Dp	0.31	0.15	0.14
7	M	Bt, Dp	0.24	0.62	0.48
8	F	Bt, Dp	0.26	0.09	0.17
9	M	Bt, Dp	0.41	0.11	0.17
10	M	Bt, Dp	0.26	0.09	0.09
11	F	Bt, Dp	0.30	0.15	0.16
12	M	Bt	0.24	0.12	0.14
13	F	Bt, Dp	0.33	0.17	0.16
14	M	Bt, Dp	0.35	0.11	0.10
15	M	Bt, Dp	0.60	0.18	0.16
16	F	Bt	1.15	0.11	0.11
17	M	Bt, Dp	0.34	0.14	0.09
18	M	Bt, Dp	0.36	0.11	0.11
19	F	Bt, Dp	0.24	0.20	0.32
20	M	Bt	0.76	0.10	0.11
21	M	Bt, Dp	0.25	0.11	0.11
22	F	Bt, Dp	0.20	0.15	0.14
23	M	Bt, Dp	0.22	0.11	0.22
24	M	Bt, Dp	0.25	0.15	0.17
25	M	Bt, Dp	0.25	0.11	0.14

Note: all patients had a positive skin prick test to house dust mites (HDM) and positive IgE to Blo t 13. Female (F), male (M), *B. tropicalis* (Bt) and *D. pteronyssinus* (Dp).

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
