# Peer review of "Identification of B Cell Epitopes of Blo t 13 Allergen and Cross-Reactivity with Human Adipocytes and Heart Fatty Acid Binding Proteins"

_ijms, 2019, doi:10.3390/ijms20246107_

Round 1

Reviewer 1 Report

Interesting work with a lot of detail. It would be nice if you had a couple tables -summarizing your findings and the clinical significance of it. You may consider expanding on the  role AIT may have in aborting or abetting autoimmune disease. There was one European claims data analysis that showed a decreased incidence of autoimmune disease in patients treated with AIT

Author Response

Review 1

Interesting work with a lot of detail. It would be nice if you had a couple tables -summarizing your findings and the clinical significance of it. You may consider expanding on the role AIT may have in aborting or abetting autoimmune disease. There was one European claims data analysis that showed a decreased incidence of autoimmune disease in patients treated with AIT.

R/: In attention to this comment, the following paragraph was added in lines 233-242

Blo t 13 is the first HDM allergen to which a cross reactivity with human proteins has been demonstrated, therefore, the probability that an autoreactive response to homologous proteins such as FABP3 and FABP4, could be induced during the course of an AIT with HDM extract deserves evaluation. In some studies, immunologic diseases such as autoimmune diseases are considered a relative contraindication for AIT [31]. However, in a large national wide study analyzing data over a decade, patients treated with AIT, had lower incidence of autoimmunity compared to those on conventional treatment (base on nasal steroids and oral antihistamines) [32]. In addition, Bozek et al [33] found in a long-term observational study that specific immunotherapy with HDM extract is indeed safe and did not find evidence to support that this immunotherapy is associated with increased risk of development of autoimmune disease.

The following references were added in the text:

31. Agache, I.; Lau, S; Akdis, CA.; Smolinska, S.; Bonini, M.; Cavkaytar, O., et al. EAACI Guidelines on Allergen Immunotherapy: House dust mite-driven allergic asthma. Allergy. 2019,74(5):855-73.

32. Linneberg A, Jacobsen RK, Jespersen L, Abildstrøm SZ. Association of subcutaneous allergen-specific immunotherapy with incidence of autoimmune disease, ischemic heart disease, and mortality. J Allergy Clin Immunol. 2012 Feb;129(2):413-9.

33. Bozek, A.; Kozlowska, R.; Jarzab, J., The safety of specific immunotherapy for patients allergic to house-dust mites and pollen in relation to the development of neoplasia and autoimmune disease: a long-term, observational case-control study. Int Arch Allergy Immunol. 2014,163(4):307-12.

Reviewer 2 Report

In this study, two IgE-binding regions were identified in Blo t 13 and cross-reactivity to FABP3 and FABP4 was investigated. The results were in general with acceptable quality but the following comments should be noted:

It is not clear why Blo t 13 was selected for epitope mapping because the allergenicity of Blo t 13 is much lower than Blo t 1, 3, 5, 11,12. For many figures, error bars and statistically analysis were absent, e.g. Figure 4, 5, 6, 7CD. Have the experiments replicated for many times? How many times have been repeated? The absence of statistical analysis will make the evaluation of results difficult. From Figure 7A, possible antigenic region(s) with peptide 7 could not be totally rule out because the peptide was still positive for at least four out of seven patients.

Author Response

Review 2

Why Blo t 13 was selected for epitope mapping because the allergenicity of Blo t 13 is much lower than Blo t 1, 3, 5, 11, 12?

R/. Blot 13 was selected because this allergen was originally characterized by our group using molecular cloning, it has structural homology with FABP family and low frequency of IgE reactivity in HDM allergy patients. However, some patients reacted with high intensity (Caraballo L, et al. Cloning and IgE binding of a recombinant allergen from the mite Blomia tropicalis, homologous with fatty acid-binding proteins. Int Arch Allergy Immunol. 1997;112:341-7). There are few studies with this allergen. In addition, regarding to the high structural homology with human proteins belonging to FABP, we hypothesized that Blo t 13 could show IgE cross reactivity with the human FABPs.

For many figures, error bars and statistically analysis were absent, e.g. Figure 4, 5, 6, 7 CD.

R/:  Error bars and statistical analysis were added to the revised version

Have the experiments replicated for many times? How many times have been repeated?

R/:  All experiments were done by duplicated.

From Figure 7A, possible antigenic region (s) with peptide 7 could not be totally rule out because the peptide was still positive for at least four out of seven patients.

R/: For Epitope mapping was defined “CRI values below 25% were considered as representing no inhibition and those between 25% and 50% as having a low degree of inhibition” (lines 362, 363). For this peptide only the serum Cam 023 (pink bar) showed CRI value between 25% and 50%. For this reason, we did not use peptide 7 further analysis.